# Long-Term Cross Immune Response in Mice following Heterologous Prime-Boost COVID-19 Vaccination with Full-Length Spike mRNA and Recombinant S1 Protein

**DOI:** 10.3390/vaccines11050963

**Published:** 2023-05-09

**Authors:** Dandan Li, Heng Zhao, Yun Liao, Guorun Jiang, Pingfang Cui, Ying Zhang, Li Yu, Shengtao Fan, Hangwen Li, Qihan Li

**Affiliations:** 1Yunnan Key Laboratory of Vaccine Research and Development on Severe Infectious Diseases, Institute of Medical Biology, Chinese Academy of Medical Sciences & Peking Union Medical College, Kunming 650118, China; lidandan@imbcams.com.cn (D.L.); zhaoheng@imbcams.com.cn (H.Z.); liaoyun@imbcams.com.cn (Y.L.); jgr@imbcams.com.cn (G.J.); cuipingfang@imbcams.com.cn (P.C.); cherryzhang629@126.com (Y.Z.); yuli@imbcams.com.cn (L.Y.); fst@imbcams.com.cn (S.F.); 2Stemirna Therapeutics Co., Ltd., Shanghai 201206, China

**Keywords:** SARS-CoV-2, heterologous vaccination, intradermal administration, intramuscular administration, immune response

## Abstract

(1) Background: As the COVID-19 pandemic enters its fourth year, it continues to cause significant morbidity and mortality worldwide. Although various vaccines have been approved and the use of homologous or heterologous boost doses is widely promoted, the impact of vaccine antigen basis, forms, dosages, and administration routes on the duration and spectrum of vaccine-induced immunity against variants remains incompletely understood. (2) Methods: In this study, we investigated the effects of combining a full-length spike mRNA vaccine with a recombinant S1 protein vaccine, using intradermal/intramuscular, homologous/heterologous, and high/low dosage immunization strategies. (3) Results: Over a period of seven months, vaccination with a mutant recombinant S1 protein vaccine based on the full-length spike mRNA vaccine maintained a broadly stable humoral immunity against the wild-type strain, a partially attenuated but broader-spectrum immunity against variant strains, and a comparable level of cellular immunity across all tested strains. Furthermore, intradermal vaccination enhanced the heterologous boosting of the protein vaccine based on the mRNA vaccine. (4) Conclusions: This study provides valuable insights into optimizing vaccination strategies to address the ongoing challenges posed by emerging SARS-CoV-2 variants.

## 1. Introduction

The COVID-19 pandemic, which began in Wuhan, China, in 2019, has now entered its fourth year. As of March 2023, the World Health Organization had reported 760 million confirmed cases and 6.87 million deaths due to SARS-CoV-2 [1]. A total of 183 vaccines have entered clinical trials (with 11 approved for market and currently in post-marketing Phase 4 clinical trials), and 199 vaccines are in pre-clinical research. Approximately 5 billion people have received two vaccine doses, while 5.5 billion have had at least one dose [1]. However, substantial research data have confirmed that six months post-vaccination, neutralizing antibody levels produced by inactivated, mRNA, and adenovirus vaccines have decreased to varying degrees. Concurrently, the ongoing prevalence and evolution of the virus have resulted in new variant strains, and the neutralizing ability of vaccines against these variants has also diminished. The emergence of the Delta and Omicron strains, in particular, has significantly challenged the protective efficacy of vaccines based on prototype strains.

Limited vaccine supply during emergencies has led some individuals initially vaccinated with mRNA vaccines to receive different vaccine types, such as adenovirus vaccines, for their second dose. Subsequent studies have shown that heterologous vaccination can produce immune effects that are not inferior to, or even superior to, homologous vaccination [2,3,4]. Moreover, studies have demonstrated that boosting with different vaccine types after primary vaccination significantly increases neutralizing antibody levels [5]. However, the duration and spectrum of protection against variants provided by heterologous prime-boost strategies require further validation. On the other hand, intradermal immunization involves injecting antigens into the dermis using a specialized intradermal injection needle. Previous research on other vaccines has shown that intradermal immunization can achieve better immune effects with smaller antigen dosages. For example, our data from inactivated vaccines against hand, foot, and mouth disease suggest that intradermal immunization can provide effective immunity in both mice and monkeys [6,7,8]. Studies on poliomyelitis have also highlighted the superior immunity provided by fractional doses of intradermal immunization [9], which is highly significant for reaching as many people as possible in vaccine-deficient areas. Consequently, the World Health Organization recommends intradermal immunization for polio vaccination as an alternative administration route to the current strategy. In the context of COVID-19 vaccines, we conducted preliminary studies on the short-term immune effects of inactivated vaccines through intradermal immunization [10,11], but the long-term immune effects induced by different antigen types as well as the cross-immunity against various variants remain unclear.

In this study, we designed a series of immunization procedures to systematically and concurrently compare the humoral and cellular immunity provided by a combination of full-length spike mRNA vaccine and recombinant S1 protein vaccine after intradermal/intramuscular, homologous/heterologous, and high/low dosage immunization. Our results demonstrate that, after seven months, vaccination with a mutant recombinant S1 protein vaccine based on the full-length spike mRNA vaccine maintained broadly stable humoral immunity against the prototype strain, partially attenuated but broader-spectrum immunity against variant strains, and a comparable level of cellular immunity across all tested strains. Additionally, intradermal vaccination promoted the heterologous boosting of protein vaccines based on mRNA vaccines. These findings will contribute to a deeper understanding of the immunological basis for the durability and breadth of vaccine protection, facilitating the optimization of vaccine design and development to address both existing and potential future strains. This knowledge will also inform the formulation of corresponding public health policies for the ongoing management of the COVID-19 pandemic.

## 2. Materials and Methods

### 2.1. Cells and Virus

The Vero cell strain WHO Vero 10-87, used to detect neutralizing antibodies, was provided by the WHO [12]. Vero cells were cultured in MEM-5% FCS medium at 37 °C in a 5% CO_2_ incubator to form monolayers for neutralization assays. CHO cells expressing the S1 protein with mutated sites were grown in RPMI 1640–8% FCS medium using a standard protocol [13], as described previously [10]. The SARS-CoV-2 prototype strain (Wuhan strain) KMS-1 (GenBank No: MT226610.1) was isolated from the Yunnan Infectious Hospital by IMB, CAMS, in January 2020, as described previously [12]. The Alpha variant (B.1.1.7, SARS-CoV-2/C-Tan-BJ202101(B1.1.7), CSTR.16698.06. NPRC 2.062100002) was gifted from China CDC, Beta variant (B.1.351, GDPCC-nCOV84, CSTR.16698.06. NPRC 2.062100001) from Guangdong CDC, Delta variant (B.1.617.2, CQ79, CSTR.16698.06. NPRC 6. CCPM-B-V-049-2105-8) from Chongqing CDC and Omicron variant (B.1.1.529 CCPM-B-V-049-2112-18) from Institute of laboratory animal sciences, CAMS&PUMC and stored at IMB, CAMS as described previously [14].

### 2.2. SARS-CoV-2 Protein Vaccine and mRNA Vaccine

The SARS-CoV-2 protein vaccine, using a recombinant S1 protein with six mutations as antigen, was constructed as described previously [10]. Briefly, the vaccine comprises a recombinant protein based on the optimized S1 sequence from the SARS-CoV-2 prototype strain, incorporating six key mutations from variants of concern: N501Y and D614G from the Alpha strain (B.1.1.7); K417N from the Beta strain (B.1.351); and L452R, E484Q, and P681R from the Delta strain (B.1.617.2). The recombinant S1 protein containing a His-tag was expressed in a CHO system, purified, and formulated at 10 μg/dose with Al(OH)_3_ adjuvant (0.0175 mg of Al/dose) for the immunological study. The SARS-CoV-2 mRNA vaccine, as described previously [15], was based on the full length of the Spike gene from the SARS-CoV-2 prototype strain and packaged in core–shell-structured lipopolyplex (LPP) nanoparticles.

### 2.3. Animal Ethical Approval

Female Balb/c mice aged 4–6 weeks were purchased from Beijing Vital River Laboratory Animal Technology Co., Ltd. (Beijing, China). The animal experiment was conducted according to the principles outlined in the “Guide for the Care and Use of Laboratory Animals” and “Guidance for Experimental Animal Welfare and Ethical Treatment”. The protocols were reviewed and approved by the Animal Experiment Ethics Committee of the Institute of Medical Biology (IMB), Chinese Academy of Medical Sciences (CAMS) (approval no.: DWSP 202,203 023). All animals were cared for by veterinarians at the IMB, CAMS.

### 2.4. Animal Immune Study

Fifty-four mice were randomly divided into nine groups, with six mice in each group, including eight experimental groups and one control group. The eight experimental groups consisted of four heterologous mRNA/protein groups (Groups A–D), two homologous mRNA groups (Group E and F), and two homologous protein groups (Group G and H). The heterologous mRNA/protein groups utilized a combination of the first dose of the mRNA vaccine and the second dose of the protein vaccine, with two mRNA dosages of 5 μg and 20 μg. The dosage for the protein vaccine was 10 μg in all corresponding groups. The administration route of vaccines included intramuscular and intradermal injections at the same location on the mice’s inner thighs, with MicronJet 600 Microneedles (NanoPass Technologies, Ltd., Hamerkaz, Israel) used for intradermal injections according to the manufacturer’s instructions. The immunization procedures for each group were as follows:

Group A/B: heterologous immunization, intramuscular injection, 5 μg/20 μg mRNA + 10 μg protein;

Group C/D: heterologous immunization, intradermal injection, 5 μg/20 μg mRNA + 10 μg protein;

Group E/F: homologous immunization, intramuscular/intradermal injection, 20 μg mRNA + 5 μg mRNA;

Group G/H: homologous immunization, intramuscular/intradermal injection, 10 μg protein + 10 μg protein.

Blood samples were collected from all mice before (T0) and 21 days after (T1) the first vaccination, and at 7 days (T2), 14 days (T3), 28 days (T4), 6 months (T5), and 7 months (T6) after the second vaccination. All mice were euthanized at T6, and PBMCs from their spleens were collected for ELISpot detection. The control group was sampled and euthanized at the same time points to collect blood samples and PBMCs, with the results used as baseline values.

### 2.5. Neutralization Assay for Wild-Type and Variant Strains of SARS-CoV-2

Serum samples were heat-inactivated at 55 °C for 30 min, then serially diluted 2-fold and co-incubated with live virus (100 lgCCID50/well) for 2 h at 37 °C. A 100 μL/well Vero cell suspension (10^5^ cells/mL) was added to the mixture in a 96-well plate, followed by incubation at 37 °C with 5% CO_2_ for 7 days. Cytopathic effects were assessed using an inverted microscope (Nikon Solutions Co., Ltd., Tokyo, Japan) to determine serum neutralizing antibody titers. Geometric mean titers (GMTs) of neutralizing antibodies were measured.

### 2.6. ELISA Assay for Wild-Type and Variant Strains of SARS-CoV-2

An ELISA assay assessing binding antibody levels against the wild-type SARS-CoV-2 was conducted using the wild-type Spike protein. ELISA plates (Corning Costar; Corning, NY, USA) were coated with S protein (Sanyou Biopharmaceuticals Co., Shanghai, China) at a concentration of 5 μg/well and incubated overnight at 4 °C. During the experiment, ELISA plates were blocked with 5% BSA-phosphate-buffered saline (PBS), incubated with serially diluted serum samples, and visualized by reaction with an HRP-conjugated antibody (Abcam, Waltham, MA, USA) and TMB substrate (Solarbio, Beijing, China), using previously described methods [16]. The absorbance of each well at 450 nm was measured using an ELISA plate reader (Gene Company, Beijing, China). Antibody serum samples yielding OD values at least 2.1-fold higher than the negative control at a test sample dilution of 1:400 were considered positive. Endpoint titers (ETs) were defined as the highest serum dilutions yielding positive OD values. GMTs were calculated as the geometric mean of the ETs of positive serum samples in each group. ELISA assays assessing binding antibody levels against variant strains of SARS-CoV-2 (Alpha, Beta, Gamma, Delta, and Omicron) were conducted using commercial ELISA kits (ACRO Biosystems Co., Beijing, China) according to the manufacturer’s instructions.

### 2.7. ELISpot Assays for Wild-Type and Variant Strains of SARS-CoV-2

ELISpot assays were performed with the Mouse IFN-γ ELISPOT Kit (Mabtech, Cincinnati, OH, USA) following the manufacturer’s instructions. Briefly, peripheral blood mononuclear cells (PBMCs) were isolated from mice’s spleens using a lymphocyte isolation technique (Ficoll-Paque PREMIUM; GE Healthcare, Piscataway, NJ, USA) and plated in duplicate wells. Recombinant Spike proteins of wild-type and variant strains of SARS-CoV-2 (Sino Biological, Inc., Beijing, China) were added to separate wells as stimulators. Phytohemagglutinin (PHA) was added as the positive control. The plates were incubated at 37 °C for 24 h, after which cells and medium were removed to allow spot development. An automated ELISPOT reader (CTL, Cleveland, OH, USA) was used to count the colored spots. Spot-forming cells (SFCs) were T cells that produced SARS-CoV-2-specific IFN-γ.

### 2.8. Statistical Analysis

Data are presented as the mean or geometric mean and standard deviation (SD). Differences among groups were evaluated by two-way analysis of variance (ANOVA) and unpaired *t* tests. GraphPad Prism software (San Diego, CA, USA) was used for statistical analyses. A *p*-value < 0.05 was considered statistically significant.

## 3. Results

### 3.1. Neutralization of SARS-CoV-2 WT Strain by Sera from S mRNA/S1 Protein Heterologous Immunization

We designed eight experimental groups (Group A–H) with two mRNA vaccine dosages (5 μg and 20 μg) and one protein vaccine dosage (10 μg) for intradermal or intramuscular injections on Day 0/21 in mice. Blood samples were collected at various time points (T0–T4, Figure 1B) to test for neutralization antibody levels against the wild-type SARS-CoV-2. 

Assessing the differences in geometric mean titers (GMT) among groups at various time points within 28 days after the second vaccination, we observed that compared to baseline (Figure 2A), high-dose mRNA intradermal groups (Groups D and F) exhibited the highest neutralizing antibody levels at T1 (21 days post-first dose), with a more than 20-fold increase, followed by intramuscular groups (Groups B and E) with a 10-fold increase (Figure 2B). The intradermal and intramuscular protein groups (Groups G and H) had the lowest increase. After the second dose, the peak level appeared at T2 (7 days post-second dose); the highest neutralizing antibody level was observed in the homologous mRNA intradermal group (Group F), followed by the homologous mRNA intramuscular group (Group E). In the high-dose heterologous groups, the intradermal group (Group D) had a higher peak level than the intramuscular group (Group B). Notably, in heterologous groups, the first low-dose mRNA dose induced relatively lower antibody levels, and the second intramuscular protein vaccine dose could effectively boost immunity, compensating for the initial low dose and achieving antibody levels comparable to the high-dose group (Groups A and B). However, with intradermal injection, the second dose had a limited boosting effect, resulting in a significant difference in final antibody levels between low- and high-dose groups (Groups C and D). The peak time of antibody level in the low-dose group (Group C) was delayed to T3 (14 days post-second dose). The lowest neutralizing antibody levels were observed in homologous protein groups (Groups G and H). After the first vaccination, there was no significant difference between the intramuscular and intradermal groups, but the intramuscular group (Group G) showed better boosting after the second vaccination, resulting in a higher overall and peak level than the intradermal group (Group H).

### 3.2. Cross-Reactive Antibodies Targeting VOCs Promoted via Boosting with Recombinant S1 Protein

We analyzed the geometric mean titers (GMT) of binding antibody levels to the WT strain at T0–T4 using ELISA (Figure 3A,B). Similar to the changes observed in neutralizing antibodies, binding antibody levels increased 5.7–25.4 fold across all eight groups at 21 days after the first vaccination. In the mRNA-initiated groups (Groups A–F), there was no significant difference between intradermal and intramuscular groups under the same dosage. However, in the protein-initiated groups (Groups G and H), the intramuscular group outperformed the intradermal group. After the second vaccination, the binding antibody levels in both the heterologous groups (Groups A–D) and the homologous mRNA groups (Groups E and F) were substantially boosted, while the homologous protein groups (Groups G and H) exhibited only a four-fold increase. Compared to the baseline level (T0), the highest fold increase, 724.1-fold, was observed in the homologous mRNA intradermal group (Group F) at 7 days after the second vaccination (T2). In the low-dose heterologous intradermal group (Group C), the peak was delayed to 28 days after the second vaccination (T4), whereas no delay was observed in the low-dose intramuscular group (Group A). The peak in Group A occurred 7 days after the second immunization (T2); however, the decline from the peak was faster, appearing 28 days after the second immunization (T4). The overall levels in the two homologous protein groups (Groups G and H) were the lowest. No significant difference was observed between these groups and others at T1; they even surpassed the low-dose heterologous intradermal group (Group C). However, after the second vaccination, these two groups fell behind the others, with the intramuscular group (Group G) outperforming the intradermal group (Group H). A peak-delay was also observed in the intradermal group (Group H), with the peak appearing 14 days after the second vaccination (T3).

To assess the cross-reactivity of the binding antibodies produced by boosting with recombinant mutant S1 protein against other variant strains, we used ELISA kits pre-coated with S proteins from different variants to analyze the sera samples from T0–T4 (Figure 3C–L). As with the WT strain, the GMT levels of binding antibodies against variant strains in all eight groups increased compared to the baseline at 21 days after the first vaccination (T1). The six mRNA-initiated groups (Groups A–F) showed a larger increase, while the two protein-initiated groups (Groups G and H) exhibited a smaller increase. At 7 days after the second vaccination (T2), the four heterologous groups (Groups A–D) displayed a lower fold increase compared to T1, and the two homologous mRNA groups (Groups E and F) showed slightly higher fold increases. The overall levels were the highest in the homologous mRNA groups (Groups E and F), with the intradermal group (Group F) outperforming the intramuscular group (Group E). In the four heterologous groups, no significant difference was observed in the high-dose groups (Groups B and D); however, in the low-dose groups, the peak time for the intradermal group (Group C) was delayed, and attenuation was accelerated in the intramuscular group (Group A). The lowest overall levels were found in the two homologous protein groups (Group G and H), in which the intramuscular group (Group G) outperformed the intradermal group (Group H). The results showed that the binding antibody levels against the Alpha, Beta, Gamma, and Delta strains in the homologous mutant protein groups (Groups G and H) were significantly lower than those in the groups initiated with WT mRNA (Groups A–F). However, this difference was not as pronounced for the Omicron strain. The mutant protein effectively increased reactivity against the Omicron strain after the first and second vaccinations, resulting in final antibody levels comparable to those in the other groups. This suggests that immunization with an antigen containing mutations from multiple variants during the first vaccination could help develop immunity that favors more diverse strains (such as Omicron) than the WT. Moreover, while GMT values indicated differences between groups, statistical analysis substantiated significance only for the most apparent comparisons (e.g., Group E and F versus Group G and H). The small sample size might affect these results, and additional studies are required for confirmation.

### 3.3. Over Seven-Month, Long-Term Antibody Response Induced by S mRNA/S1 Protein Heterologous Immunization

To monitor the long-term antibody response induced by different immunization procedures, we collected blood samples at six months (T5) and seven months (T6) after the second vaccination to analyze antibody levels in the sera. The neutralization test results for T5 and T6 sera revealed that the neutralization ability against the WT strain remained stable for all groups after seven months (Figure 4A), with less than a two-fold change compared to 28 days post-second vaccination (T4, Figure 4B). Regarding binding antibody levels in the ELISA, reactivity against the WT S protein remained stable in groups initiated with mRNA (Group A–F), with a change of less than two-fold (Figure 4C,I). There was even a small increase in the high-dose heterologous intradermal group (Group B). However, homologous protein groups experienced a 2–4-fold decrease compared to T4. We then assessed the cross-reactivity of each group against other variants (Figure 4D–I). All groups displayed varying degrees of decrease, with more severe reductions observed for Gamma, Delta, and Omicron variants, showing an up-to-eight-fold decrease against Omicron. Among these, the low-dose heterologous intradermal group (Group C), which had a delayed peak time, exhibited a more severe decrease in binding antibody levels against all variants. The two homologous protein groups (Groups G and H) also experienced severe decreases, even with lower overall antibody levels.

Seven months after the second vaccination (T6), we tested the sera’s neutralizing ability against variant strains and compared it to the neutralizing ability against the WT strain (Figure 5A,B). Overall, the GMT levels of neutralizing antibodies were higher in the groups with mRNA initiation (Groups A–F) and lowest in the homologous protein groups (Groups G and H). Within the heterologous intramuscular groups (Groups A and B), the high-dose group (Group B) demonstrated higher neutralization ability against multiple variants, significantly differing from the low-dose group (Group A). However, in the heterologous intradermal groups (Groups C and D), the low-dose group (Group C) exhibited a neutralization ability comparable to that of the high-dose group (Group D). This advantage of intradermal immunization was also observed in the two homologous mRNA groups (Groups E and F), corresponding to the highest overall antibody levels in the homologous mRNA intradermal group (Group F), as previously described. Although the two homologous protein groups (Groups G and H) retained a certain level of binding antibody, the neutralizing antibody levels were extremely low. Interestingly, after seven months, the high-dose heterologous intramuscular group (Group B), both high- and low-dose heterologous intradermal groups (Groups C and D), and the homologous mRNA intramuscular group (Group E) all exhibited increased neutralizing ability against the Alpha, Beta, and Delta strains compared to the WT strain, with the most pronounced increase observed for the Alpha strain (Figure 5B). The neutralizing ability against the Omicron strain was reduced by tens of folds compared to the WT strain in all groups, with the smallest reduction being only 11-fold in the high-dose heterologous intradermal group (Group D). Analogous to the previous result in binding antibody level, statistical analysis on neutralizing antibodies revealed significance only for the most noticeable comparisons (e.g., Groups E and F versus Groups G and H), and the differences within Groups A–F were not always consistently statistically significant. Further studies with larger sample sizes are necessary to confirm the conclusions.

### 3.4. Over Seven-Month, Long-Term Cellular Immune Responses Induced by S mRNA/S1 Protein Heterologous Vaccine

At the experimental endpoint of seven months post-second vaccination (T6), we sacrificed all groups of mice, collected PBMC cells from their spleens, and used variant S proteins as stimulators to assess T cell ability to secrete IFN-γ, a measure of cellular immunity. Although T cell responses displayed a trend generally similar to antibody levels—that is, cellular immune levels were generally higher in the four heterologous groups (Group A–D) and the two homologous mRNA groups (Group E and F), while the two homologous protein groups (Group G and H) exhibited the lowest levels—this group difference was not as pronounced as that observed for antibody levels (Figure 5C–E). Intriguingly, cellular immunity levels against multiple variants within each group displayed no substantial differences and were approximately the same: excluding the two homologous protein groups (Group G and H) with the lowest levels, the differences in cellular immunity levels against variants in all other groups were within a two-fold range compared to the WT strain (Figure 5D). This observation suggests that, to some extent, although the antibodies produced by vaccination might not effectively neutralize emerging variants, cellular immunity maintains a broader and more effective response for a longer period than antibody immunity.

## 4. Discussion

Although existing studies have shown that neutralizing antibody levels produced by vaccination or COVID-19 infection decrease over time, several investigations on the duration of antibody responses suggest long-term general humoral immunity can be observed up to 8 to 12 months after infection in convalescent patients [17,18]. The number of SARS-CoV-2 antigen-specific memory B cells remains stable for at least 6 to 12 months [18], and B-cell clonal selection and accumulation accompanied by neutralizing antibody release [19] support the persistence of humoral immunity after infection [20]. In terms of vaccine-induced immunity, Moderna studies have demonstrated that clinical trial participants maintained high levels of antibodies within 6 months after receiving the second dose of their mRNA-1273 vaccine [21]. Similarly, our results reveal that at seven months post-vaccination, all mouse groups that initiated with mRNA vaccines (heterologous mRNA/protein intramuscular/intradermal groups and homologous mRNA intramuscular/intradermal groups) retained stable binding antibodies, neutralizing antibody levels, and T cell responses against the WT strain.

Regarding cross-reactivity, our results showed that all mouse groups that initiated with mRNA vaccines maintained stable binding antibody levels against variant strains other than Omicron seven months after vaccination. Upon the first vaccination of wild-type S mRNA vaccine, the heterologous boost by recombinant S1 protein containing key mutations from VOCs produced an even greater neutralization ability against variants than that of the WT strain after seven months. However, due to the absence of Omicron mutation sites in the recombinant S1 protein, the neutralization ability against Omicron decreased significantly. Interestingly, the homologous protein groups that initiated with recombinant S1, despite the severe decrease in antibody and cellular immunity levels after seven months, exhibited relatively higher and more comparable binding antibody levels against Omicron in the short term, as opposed to the significant differences observed with other groups against other variant strains. Similar to our observation, in naive animals vaccinated and boosted with Omicron-based mRNA vaccines, the neutralization responses were mainly directed against the Omicron strain instead of past variants, demonstrating the “imprinting” effect of the first vaccination antigen on the immune system [16]. Therefore, the combination of a vaccine capable of cross-reacting with past variants and a specific vaccine against the current variant is more likely to ensure a greater neutralization spectrum in naive hosts [22].

In terms of cellular immunity, although currently known variants, including Beta and Delta, can reduce the neutralizing antibody levels formed by vaccination or infection, evidence has shown that these variants rarely escape memory T-cell responses [23]. The analysis of T cell epitopes also demonstrated that the T cell epitopes in the Omicron strain were largely conserved compared with the WT strain, suggesting that memory T cells may provide protection in reinfection or breakthrough infection of Omicron [24]. Our results suggest that, seven months after vaccination, effective immune responses were established by homologous or heterologous vaccination initiated with the WT S mRNA vaccine. Although the differences in neutralization ability against variants were significant, the differences in T cell response levels against variants were not pronounced, even for Omicron, which had the lowest levels of neutralizing antibodies, but also detected significant and comparable T-cell responses. This result confirms the persistence and broad spectrum of T-cell responses.

Regarding dosage, one study shows that after two doses of low-dose mRNA vaccination in primates, most animals displayed a limited immune response [25]. Mateus et al. reported the results of a clinical trial, showing that a low dosage of Moderna mRNA vaccine can induce long-lasting neutralizing antibodies as well as memory CD4+ T cell immunity, including T follicular helper (TfH) and IFN-γ-secreting cells [26]. In our study, we designed a procedure of low-dose versus high-dose mRNA vaccination followed by the same dosage of protein vaccination. After comparing intradermal and intramuscular administration routes, we found that although the peak antibody levels in the two low-dose mRNA groups were similar, the peak time in the intradermal group was delayed to 28 days after the second vaccination, whereas no such delay was observed in the low-dose intramuscular mRNA group. In addition, at the end of seven months, the low-dose intradermal group exhibited the same or slightly higher neutralizing antibody levels as the high-dose group. This seems to indicate that low-dose intradermal heterologous immunization induces a slower increase, but ultimately long-lasting and broad-spectrum immunity. Furthermore, after high-dose mRNA vaccination, the boosting of a low-dose mRNA was effective enough to enhance immunity, and this high-low-dose prime-boost procedure performed better via the intradermal route.

Currently, there is substantial evidence supporting that heterologous immunization can provide effective boosting; it has also been shown that mRNA and protein heterologous vaccination can improve the broad spectrum of produced antibodies [27]. Our study demonstrated that although the homologous mRNA groups exhibited the highest peak level of antibodies, after a slow decay of seven months, antibody levels in the heterologous groups reached a level more comparable to that of the homologous mRNA groups, particularly evident when tested against the Delta and Omicron strains. In the homologous mRNA groups, the broad-spectrum was more apparent in the intradermal group than in the intramuscular group, and the intradermal route compensated to some extent for the limits on the broadness of the spectrum provided by homologous mRNA vaccination. This result suggests that intradermal and intramuscular routes appear to have inconsistent kinetics and mechanisms of action. In the homologous mRNA groups, under the same dosage, the effect of the intradermal route was superior to that of the intramuscular route. In heterologous groups, the dosage of the first mRNA had a stronger impact on the ultimate antibody level in the intradermal route than the intramuscular route. In the homologous protein groups, the intramuscular group performed better overall than the intradermal group. Taken together, these results suggest that mRNA could play a more effective role in the intradermal route, and the high dosage of the first vaccination could significantly increase the antibody level and the speed of antibody production during the entire immune process. Moreover, the protein can play a more effective role in the intramuscular route. The intradermal administration route has been used in many vaccines [28], and there is evidence that it can elicit the same level of protection with a reduced amount of antigen [9,29], which is economically important for the goal of extending as many vaccines as possible to less developed areas in an emergency to achieve ultimate herd immunity. However, the immune mechanisms underlying the intradermal and intramuscular routes are still largely unknown, and further exploration on this topic will help to understand how the immune system responds to antigens from different routes and to formulate more targeted vaccine administration strategies corresponding to different types of antigens.

Although the global COVID-19 situation is currently well controlled three years into the pandemic due to widespread vaccination efforts and increased public health awareness, the potential for new emerging variants remains a concern. Additionally, a substantial number of individuals have received incomplete or no vaccination with existing vaccines because of limited healthcare access or medical contraindications. These factors, coupled with the threat of waning immunity over time, underscore the importance of developing novel vaccines or optimizing current vaccine strategies, as explored in this study, to maintain long-term control over the pandemic. One distinctive aspect of our vaccine strategy is its adaptability to emerging variants. By initially vaccinating with a prototype mRNA vaccine and subsequently administering a boost with a vaccine based on newly emerged variants, our approach aims to provide durable and flexible immunity, which could potentially keep the immune response in sync with the evolving viral landscape. Going forward, our intention is to focus on refining the formulation and optimizing the vaccination strategy, including factors such as administration route, combination of different types, and dosage, as considered in the design of our vaccination procedures. In light of existing widespread vaccination efforts, our proposed clinical trial will concentrate on specific populations that may benefit most from our vaccine strategy. For example, the trial could target individuals with suboptimal immunity due to incomplete vaccination, waning immunity, or a poor response to the initial vaccine. Additionally, the trial could enroll individuals who have not yet been vaccinated due to medical contraindications or other reasons, offering a potential alternative for these populations.

A limitation of our study is the small sample size, with six mice per group. The differences between Groups A–F and the homologous protein groups (G and H) are significant, while the differences among the six groups often lack statistical significance, despite the observed differences in GMT values. Although the geometric mean titer (GMT) of antibody levels is a widely accepted parameter for comparing immunological effects in vaccine studies because it accounts for the log-normal distribution of antibody titers often observed in immunological data and provides a more accurate summary of central tendency, the small sample size in our study may limit the statistical power to detect significant differences in antibody levels between groups, particularly in Groups A–F. Further studies with larger sample sizes are needed to corroborate our findings and provide stronger evidence for the immunological effects of different vaccination procedures.

## Figures and Tables

**Figure 1 vaccines-11-00963-f001:**
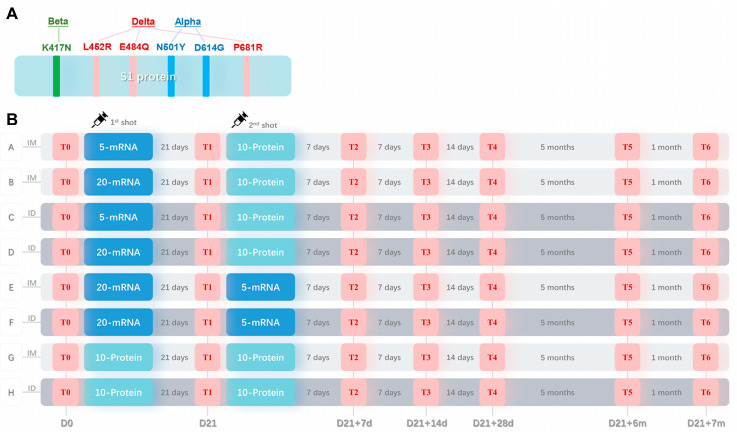
Sequence design for the recombinant S1 protein and the mouse vaccination procedure. (**A**) An optimized sequence of the S1 protein with six mutated sites (N501Y, K417N, E484Q, L452R, P681R, and D614G) derived from three VOCs: Alpha, Beta, and Delta. (**B**) Animal immunization schedule. Mice were randomly divided into eight experimental groups (Groups A–H) and one control group (not shown) (n = 6). Groups A, B, E, and G received intramuscular injections (IM, as shown in the light gray shadow), and Groups C, D, F, and H received intradermal injections (ID, as shown in the dark gray shadow) twice, as indicated by syringe symbols. 5-mRNA or 20-mRNA, 5 μg or 20 μg of S mRNA vaccine (as shown in the dark blue shadow); 10-Protein, 10 μg of recombinant S1 protein vaccine (as shown in the light blue shadow). T0–T6, six time points of blood sample collection (as shown in the red shadow); the interval time between each time point is shown in gray letters, and the number of days from the first injection is shown in gray letters below.

**Figure 2 vaccines-11-00963-f002:**
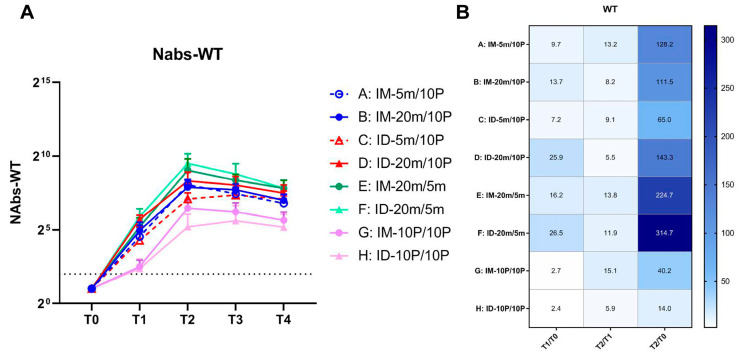
Neutralizing antibody levels against wild-type SARS-CoV-2 within 28 days after the second vaccination. (**A**) Neutralizing antibodies induced by 5 m (5 μg of S mRNA)/20 m (20 μg of S mRNA)/10 P (10 μg of recombinant S1 protein) injections via IM (intramuscular) or ID (intradermal) routes. Blood samples were collected from Groups A–H of mice before (T0) and 21 days after (T1) the first vaccination, and at 7 days (T2), 14 days (T3), 28 days (T4) after the second vaccination. The neutralizing antibodies of each group at every time point are shown as GMTs (Geometric mean titers) with bars (SD). Statistical significance was assessed by two-way ANOVA. F versus A, B, C, G, H: ****, C: ***; E versus G, H: ****, C: ***, A: **, B: *; B versus H: *; D versus H: ***, G: **. (*, *p* < 0.05, **, *p* < 0.01, ***, *p* < 0.001, ****, *p* < 0.0001). (**B**) Fold changes from T0 to T1, T1 to T2, and T0 to T2 were calculated using the GMTs of each time point. Fold change = 1 (as shown in white) represents no changes between these two time points, fold change > 1 (as shown in blue) represents an increase from the first time point to the second time point.

**Figure 3 vaccines-11-00963-f003:**
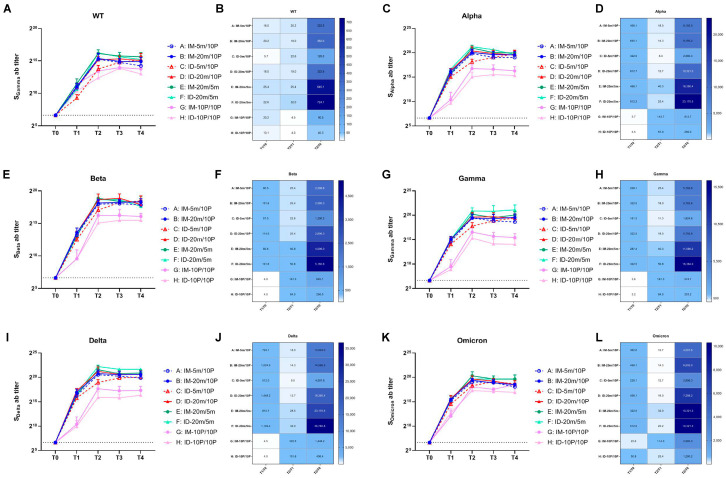
Binding antibody levels against wild-type and variant strains of SARS-CoV-2 within 28 days after the second vaccination. (**A**,**C**,**E**,**G**,**I**,**K**) Binding antibodies induced by 5 m (5 μg of S mRNA)/20 m (20 μg of S mRNA)/10 P (10 μg of recombinant S1 protein) injections via IM (intramuscular) or ID (intradermal) routes. Blood samples were collected from Groups A–H of mice before (T0) and 21 days after (T1) the first vaccination, and at 7 days (T2), 14 days (T3), 28 days (T4) after the second vaccination. The binding antibodies of each group at every time point were tested with ELISA and are shown as GMTs (Geometric mean titers) with bars (SD). Statistical significance was assessed by two-way ANOVA. (**A**) F versus A, C, G, H: ****, B: ***, D: **; E versus G, H: ****, A, C: ***, B, D: *; B versus H: *; D versus G: *, H: **; (**C**) F versus A, B, C, D, G, H: ****, E: *; E versus G, H: ****, A, C: **; A versus H: *; B versus G: **, H: ***; D versus G, H: ****. (**E**) F versus G, H: ****, A, C: ***, B: **; E versus G, H: **; B versus H: *; D versus G, H: **. (**G**) F versus A, C, G, H: ****, B, D: ***, E:**; E versus G, H: **. (**I**) F versus A, B, C, G, H: ****, D: ***, E: *; E versus G, H: ****, C: **; B versus G, H: **; D versus G, H: ***. (**K**) F versus G, H: ****, C: ***, A: **, B: *; E versus G, H: ****, A, C: ***, B: **, D: *; B versus H: *; D versus H: **. (*, *p* < 0.05, **, *p* < 0.01, ***, *p* < 0.001, ****, *p* < 0.0001). (**B**,**D**,**F**,**H**,**J**,**L**) Fold changes from T0 to T1, T1 to T2, and T0 to T2 were calculated using the GMTs of each time point. Fold change = 1 (as shown in white) represents no changes between these two time points, fold change > 1 (as shown in blue) indicates an increase from the first time point to the second time point.

**Figure 4 vaccines-11-00963-f004:**
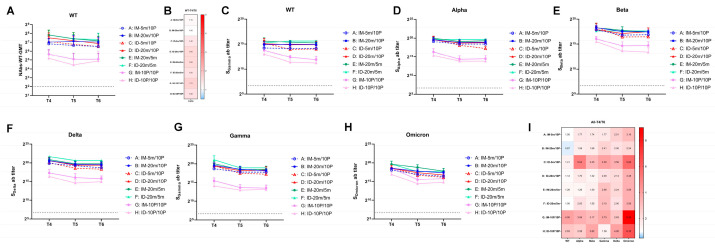
Over seven-month, long-term neutralization and binding antibody responses against wild-type and variant strains of SARS-CoV-2. (**A**,**C**–**H**) Neutralizing antibodies against wild-type, binding antibodies against wild-type, and variant strains induced by 5 m (5 μg of S mRNA)/20 m (20 μg of S mRNA)/10 P (10 μg of recombinant S1 protein) injections via IM (intramuscular) or ID (intradermal) routes. Blood samples were collected from Groups A–H of mice at 28 days (T4), 6 months (T5), and 7 months (T6) after the second vaccination. The neutralizing antibodies of each group at every time point are shown as GMTs (Geometric mean titers) with bars (SD). Statistical significance was assessed by two-way ANOVA. (**A**) F versus G, H: ****, A: **, C: *; E versus G, H: ****, A: **, C: *; A versus H: *; B versus G: ***, H: ****; C versus G: *, H: **; D versus G, H: ****. (**C**) F versus A, G, H: ****, C: ***, B, D: *; E versus G, H: ****, A: ***, C: *; B versus G: **, H: ***; D versus G: **, H: ***; (**D**) F versus G, H: ****, A,C: **, D: *; E versus G, H: ****; B versus G, H: **; D versus G: *, H: **. (**E**) F versus G, H: ****; E versus G, H: *; B versus G, H: *. (**F**) F versus G, H: ****, A: **, C, D:*. (**G**) F versus A, B, C, D, G, H: ****, E: **; E versus G, H: ****; B versus G: **, H: ***; D versus G, H: **. (**H**) F versus H: ****, A, G: ***, C: **, D: *; E versus H: ****, A, G: ***, C, D: **, B: *. (*, *p* < 0.05, **, *p* < 0.01, ***, *p* < 0.001, ****, *p* < 0.0001). (**B**,**I**) Fold changes from T4 to T6 were calculated using the GMTs of each time point. Fold change = 1 (as shown in white) represents no changes between these two time points, fold change > 1 (as shown in red) indicates a decrease from T4 to T6, fold change < 1 (as shown in blue) signifies an increase from T4 to T6.

**Figure 5 vaccines-11-00963-f005:**
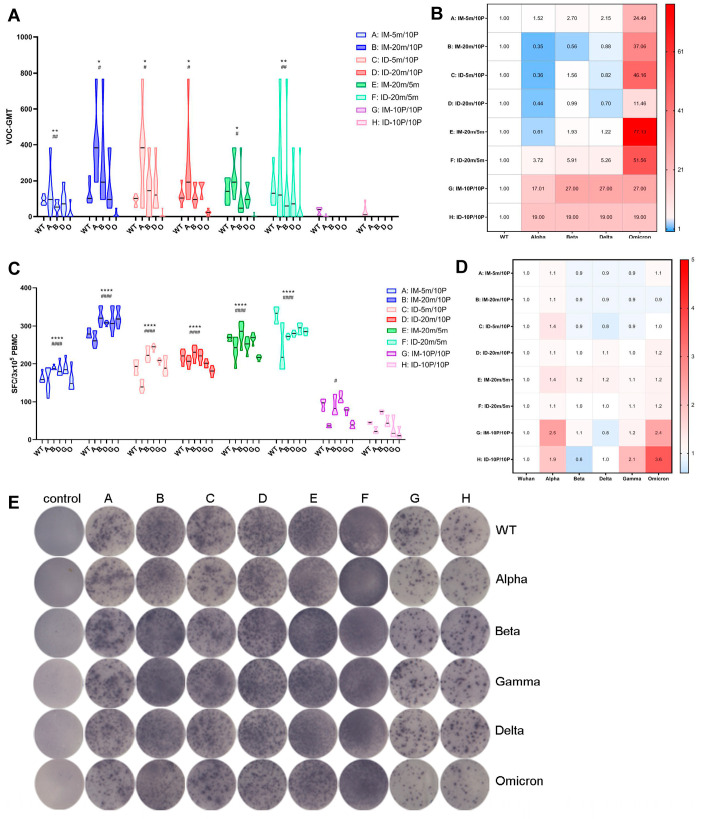
Over seven-month, long-term antibody and cellular responses against wild-type and variant strains of SARS-CoV-2. (**A**,**C**) Neutralizing antibodies and specific T cell responses against wild-type and variant strains of SARS-CoV-2 at T6, with each point representing one mouse in the group. WT, wild-type; A, Alpha; B, Beta; D, Delta; G, Gamma; O, Omicron. Statistical significance was assessed by the unpaired *t* test. *, *p* < 0.05, **, *p* < 0.01, ****, *p* < 0.0001 versus G; #, *p* < 0.05, ##, *p* < 0.01, ####, *p* < 0.0001 versus H. (**B**,**D**) Fold differences in neutralizing antibody levels and specific T cell responses against variant strains compared to wild-type. Fold difference = 1 (as shown in white) represents no difference between wild-type and VOC, fold difference > 1 (as shown in red) indicates a reduced VOC level compared to wild-type, fold difference < 1 (as shown in blue) denotes an increased VOC level compared to wild-type. (**E**) Representative spot diagrams of the ELISPOT analysis for IFN-γ-secreting T cell responses elicited by wild-type and variant strains of the Spike protein at T6. The control group was sampled and euthanized at the same time points as the vaccination groups, as described in the Section 2. Samples were run in duplicate.

## Data Availability

Not applicable.

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
