# Peer review of "Long-Term Cross Immune Response in Mice following Heterologous Prime-Boost COVID-19 Vaccination with Full-Length Spike mRNA and Recombinant S1 Protein"

_vaccines, 2023, doi:10.3390/vaccines11050963_

Round 1
Reviewer 1 Report
Manuscript is well written and showed humoral and cellular immune response. However, it seems like COVID situation is in a good control now all over the world due to multiple vaccinations, sanitation and individual and cumulative awareness. What the authors think to go forward with this current pre-clinical vaccine?
What is the ambition regarding this vaccine?
What is the plan for the investigators to run clinical trial by using this current vaccine?
These points need to be discussed in the discussion section.
Regarding IFN-g ELISPOT, it will be good to have a photo of spots before and after vaccination.
Thank you.
Investigators have written the manuscript nicely. Few edits may be required. eg: IFN-g. "g" should be given as symbol.
Author Response
Comments and Suggestions for Authors
Manuscript is well written and showed humoral and cellular immune response. However, it seems like COVID situation is in a good control now all over the world due to multiple vaccinations, sanitation and individual and cumulative awareness. What the authors think to go forward with this current pre-clinical vaccine?
Thank you for your insightful comments and suggestions. We appreciate the recognition of our manuscript's quality and presentation of humoral and cellular immune response data. Indeed, the COVID-19 situation has improved worldwide now, however, we do believe that it remains essential to continue developing and optimizing vaccination strategies to address the ongoing challenges posed by new emerging SARS-CoV-2 variants and the waning immunity. By initially vaccinating with a prototype mRNA vaccine and subsequently administering a boost with a vaccine based on newly emerged variants, our approach aims to provide durable and flexible immunity, which could potentially keep the immune response in sync with the evolving viral landscape. Going forward, our intention is to focus on refining the formulation and optimizing the vaccination strategy, taking into account factors such as vaccine antigen basis, forms, dosages, and administration routes, as considered in the design of our vaccination procedures. We believe that our study provides valuable insights that will help optimize vaccine design and development, ensuring preparedness for potential future challenges and contributing to the ongoing management of the COVID-19 pandemic.
What is the ambition regarding this vaccine?
The ambition regarding this pre-clinical vaccine is to provide an additional layer of protection against COVID-19 and its variants by harnessing the benefits of a combined full-length spike mRNA and recombinant S1 protein vaccine approach. This strategy aims to enhance the immune response and broaden the spectrum of immunity against different viral strains, potentially increasing the overall effectiveness of the vaccine and reducing the risk of future outbreaks.
What is the plan for the investigators to run clinical trial by using this current vaccine?
In light of existing widespread vaccination efforts, our proposed clinical trial will concentrate on specific populations that may benefit most from our vaccine strategy. For example, the trial could target individuals with suboptimal immunity due to incomplete vaccination, waning immunity, or poor response to the initial vaccine. Additionally, the trial could enroll individuals who have not yet been vaccinated due to medical contraindica-tions or other reasons, offering a potential alternative for these populations.
These points need to be discussed in the discussion section.
In response to your comment, we have expanded our discussion section to highlight how our research findings could inform the formulation of corresponding public health policies for the ongoing management of the COVID-19 pandemic. We believe that our study provides valuable insights that will help optimize vaccine design and development, ensuring preparedness for potential future challenges.
Regarding IFN-g ELISPOT, it will be good to have a photo of spots before and after vaccination.
We appreciate the suggestion and have now added a figure (Figure 5E) showing representative IFN-g ELISPOT spot photos of all groups. For the pre-vaccination comparison, we have included the control group, which received no vaccination and was stimulated using the same stimulator as the other groups. This serves as an appropriate blank control, representing the situation before vaccination. We believe that this additional visual representation will further enhance the clarity and understanding of our results.
Comments on the Quality of English Language
Investigators have written the manuscript nicely. Few edits may be required. eg: IFN-g. "g" should be given as symbol.
Thank you for your kind words and attention to detail. We have now made the necessary edits, including changing IFN-g to IFN-γ, as suggested. Additionally, we have made other revisions throughout the manuscript, which are marked in red characters for easy identification.
Reviewer 2 Report
Li et al. conducted an evaluation of the immunogenicity of COVID-19 mRNA and protein vaccines in mice with different vaccination regimens, doses, and routes, as well as the waning of humoral and cellular responses over 7 months. The authors concluded that high-dose homologous prime-boost mRNA vaccine appeared to be superior to mRNA/protein heterologous vaccination, and mRNA vaccine maintained a broader-spectrum immunity against different variants, albeit with attenuated antibody titers. However, this manuscript has some issues that need to be addressed.
My main concern is whether the authors performed statistical analysis on the humoral and cellular responses after vaccination. Based on the plots, there doesn't seem to be a significant difference between the groups, particularly Group A-F. It's unclear if the conclusion drawn in the manuscript is conclusive or not. The authors should provide the p-value between groups in each figure to address this issue.
Minor issues:
Line 80, duplicate period
Line 88, Please specify the database for the accession number MT226610.1
Line 89, Please provide the database accession number of the sequences of the variants used in the manuscript. Did the authors perform cross-neutralization assays with the listed variants?
Line146, It should be 105 instead of 105.
Line 208, high-dose should replace "high-dosage" in the entire MS.
Figure 4, please provide a figure with higher resolution.
The English quality of the manuscript is acceptable, but it would benefit from revision by a native English speaker or a scientific editor.
Author Response
Comments and Suggestions for Authors
Li et al. conducted an evaluation of the immunogenicity of COVID-19 mRNA and protein vaccines in mice with different vaccination regimens, doses, and routes, as well as the waning of humoral and cellular responses over 7 months. The authors concluded that high-dose homologous prime-boost mRNA vaccine appeared to be superior to mRNA/protein heterologous vaccination, and mRNA vaccine maintained a broader-spectrum immunity against different variants, albeit with attenuated antibody titers. However, this manuscript has some issues that need to be addressed.
My main concern is whether the authors performed statistical analysis on the humoral and cellular responses after vaccination. Based on the plots, there doesn't seem to be a significant difference between the groups, particularly Group A-F. It's unclear if the conclusion drawn in the manuscript is conclusive or not. The authors should provide the p-value between groups in each figure to address this issue.
Thank you for your insightful comments and concerns. We understand the importance of providing statistical analysis to support our conclusions, and we have indeed performed such analyses for the humoral and cellular responses after vaccination. We have used geometric mean titers (GMTs) to measure the central tendency of the antibody titers, which is appropriate for the analysis of immunogenicity data, however, due to the small sample size in each group, the results may not reveal significant differences in some cases, especially within the Group A-F. However, we believe that the observed trends in the data are still valuable for generating insights into the different vaccination strategies.
In response to your suggestion, we have now included p-values between groups in each figure to better illustrate the statistical significance of our findings. Additionally, we have added a separate paragraph in the discussion section to address the limitations of our study, including the small sample size and the potential impact on the statistical analysis.
We hope that these revisions and clarifications adequately address your concerns and provide a more comprehensive understanding of our findings.
Minor issues:
Line 80, duplicate period
Thank you for your attention to detail. We have edited it as suggested.
Line 88, Please specify the database for the accession number MT226610.1
Thank you for your suggestion, we have added the database “GenBank” and marked in red characters for easy identification.
Line 89, Please provide the database accession number of the sequences of the variants used in the manuscript. Did the authors perform cross-neutralization assays with the listed variants?
Thank you for your suggestion, we have now added the details of the variants we used in the manuscript and marked in red characters for easy identification. And we did perform a cross-neutralization of the mice sera at 7 months after vaccination against these listed variants, as we described in Result 3.4 and illustrated in Figure 5. We hope that these additions help clarify our methods and results.
Line146, It should be 105 instead of 105.
Thank you for your attention to detail. We have edited it as suggested.
Line 208, high-dose should replace "high-dosage" in the entire MS.
Thank you for your suggestion, we have replaced "high-dosage" with high-dose, and "low-dosage" with low-dose in the entire manuscript.
Figure 4, please provide a figure with higher resolution.
Thank you for your suggestion, we have replaced the original figure with a new one with higher resolution.
Comments on the Quality of English Language
The English quality of the manuscript is acceptable, but it would benefit from revision by a native English speaker or a scientific editor.
Thank you for your suggestion. We have now had the manuscript revised by a native English speaker to improve its overall quality. The edited portions have been marked in red characters for easy identification. We appreciate your attention to detail and hope that these revisions enhance the readability and clarity of our work.
Round 2
Reviewer 2 Report
The authors have addressed all my concerns.